



**Groundwater salinity variation in Upazila Assasuni**
**(southwestern Bangladesh), as steered by surface clay layer**
**thickness, relative elevation and present-day land use**
Floris Loys Naus[1], Paul Schot[1], Koos Groen[2], Kazi. Matin Ahmed[3], Jasper Griffioen[1,4]
[1] Copernicus Institute, Environmental Sciences, Utrecht University, Utrecht, The Netherlands
[2] Acacia water, Gouda, The Netherlands
[3] Department of Geology, Dhaka University, Dhaka, Bangladesh
[4] TNO Geological Survey of the Netherlands, Utrecht, The Netherlands
*Correspondence to*: Floris L. Naus (f.l.naus@uu.nl)
**Abstract.** In the southwestern coastal region of Bangladesh, options for drinking water are limited by groundwater
salinity. To protect and improve the drinking water supply, the large variation in groundwater salinity needs to be
better understood. This study identifies the palaeo and present-day hydrological processes and their geographical
or geological controls that determine variation in groundwater salinity in Upazila Assasuni in southwestern
Bangladesh. Our approach involved three steps: a geological reconstruction, based on the literature; fieldwork to
collect high density hydrological and lithological data; and data processing to link the collected data to the
geological reconstruction in order to infer the evolution of the groundwater salinity in the study area. Groundwater
freshening and salinization patterns were deduced using PHREEQC cation exchange simulations and isotope data
was used to derive relevant hydrological processes and water sources. We found that the factor steering the relative
importance of palaeo and present-day hydrogeological conditions was the thickness of the Holocene surface clay
layer. The groundwater in aquifers under thick surface clay layers is controlled by the palaeohydrological
conditions prevailing when the aquifers were buried. The groundwater in aquifers under thin surface clay layers is
affected by present-day processes, which vary depending on present-day surface elevation. Slightly higher-lying
areas are recharged by rain and rainfed ponds and therefore have fresh groundwater at shallow depth. In contrast,
the lower-lying areas with a thin surface clay layer have brackish–saline groundwater at shallow depth because of
flooding by marine-influenced water, subsequent infiltration and salinization. Recently, aquaculture ponds in areas
with a thin surface clay layer have increased the salinity in the underlying shallow aquifers. We hypothesize that
to understand and predict shallow groundwater salinity variation in southwestern Bangladesh, the relative elevation
and land use can be used as a first estimate in areas with a thin surface clay layer, while knowledge of
palaeohydrogeological conditions is needed in areas with a thick surface clay layer.
**1 Introduction**
In the Ganges–Brahmaputra–Meghna (GBM) river delta, home to 170 million people, availability of safe drinking
water is problematic because of the very seasonal rainfall, the likelihood of arsenic occurrence in the shallow
groundwater and the pollution of surface water bodies (Harvey et al., 2002; Ravenscroft et al., 2005; Chowdhury,
2010; Sharma et al., 2010; Bhuiyan et al., 2011). In the southwestern coastal region of Bangladesh, suitable
drinking water options are even more limited, as here the groundwater is largely brackish to saline (Bahar and
Reza, 2010; George, 2013; Fakhruddin & Rahman, 2014; Worland et al., 2015; Ayers et al., 2016). Consequently,





the people in this region are at high risk of preeclampsia, eclampsia and gestational hypertension from drinking groundwater, and at increased risk of ingesting pathogens from water from the traditional ponds (Kräzlin, 2000; Khan et al., 2014). Stress on the limited reserves of fresh groundwater is expected to rise in the future through a combination of climate change, sea level rise, increased abstraction for irrigation and industry, and population growth (Shameem et al., 2014; Auerbach et al., 2015). To protect and improve the drinking water supply in the coastal region it is therefore important to understanding the present-day spatial variation and formation processes of the groundwater salinity.

Previous studies have found great variation in the groundwater salinity in southwestern Bangladesh (BGS and DHPE, 2001; George, 2013; Worland et al., 2015; Ayers et al., 2016). Several explanations for this large variation have been proposed. One is present-day saline water recharge from the tidal rivers and creeks and the aquaculture ponds that cover much of the region (Rahman et al., 2000; Bahar et al., 2010; Paul et al., 2011; Ayers et al., 2016). Another is freshwater recharge where the clay cover is relatively thin and from rainfed inland water bodies (George, 2013; Worland et al., 2015; Ayers et al., 2016). Finally, some of the variation in the salinity of the groundwater is thought to reflect historical conditions prevailing when the aquifer was buried (George, 2013; Worland et al., 2015; Ayers et al., 2016). Studies in coastal deltas elsewhere, where a higher head due to freshwater infiltration in the higher areas leads to the formation of freshwater lenses, have identified elevation differences as being important factors controlling  groundwater salinity (Stuyfzand, 1993; Walraevens et al., 2007; Goes et al., 2009; de Louw et al., 2011; Santos et al. 2012). It has been suggested that both present-day and palaeohydrological processes are important, as deltas are almost never in equilibrium with present-day boundary conditions (Sukhija et al., 1996; Groen et al., 2000; Post and Kooi, 2003; Sivan et al., 2005; Delsman et al., 2014).

It remains unclear how each of the proposed processes influences groundwater salinity variation in southwestern Bangladesh. Previous studies found no spatial autocorrelation in groundwater salinity, presumably because the sampling distances were larger than the expected variation in groundwater salinity (Ayers et al., 2016). In our study, we set out to elucidate the hydrological processes that determine the salinity variation in the groundwater by using high density sampling in a case study area with large variation in land use, surface water bodies and surface elevation. In addition, we aimed to identify geographical or geological factors controlling the dominant salinization and freshening processes and, therefore, the groundwater salinity.

## 2 Methods

### 2.1 Methodological approach

Our approach consisted of three steps. First, based on the literature, we reconstructed the geological evolution of southwestern Bangladesh. Second, in the field we carried out high density hydrological and lithological data collection to capture the large expected variation in groundwater salinity. Third, we inferred the evolution of the groundwater salinity by interpreting the field data in the light of the regional geological reconstruction and present-day surface conditions, to determine the dominant processes responsible for the variation in groundwater salinity.

### 2.2 Fieldwork

To research all the proposed processes, the study area needed to have much variation in land use and surface water bodies, and appreciable variation in surface elevation. We selected an area in southwest Bangladesh by analysing



satellite imagery (World Imagery, ESRI, Redlands, CA, USA) to distinguish land use and surface water bodies,
Shuttle Radar Topography Mission data (SRTM) (Farr et al., 2007) to analyse elevation patterns and a soil map
(FAO, 1959) to ascertain surface geology and geomorphology. The case study area, which is in the Assasuni
Upazila (Figure 1), comprises settlements on slightly higher land, surrounded by lower-lying agricultural fields
and aquaculture ponds (Figure 1). There are freshwater ponds in the settlements. The soil in the study area is
composed of fluvial silts in the higher areas and tidal flat clays in the surrounding lower-lying areas (FAO, 1959).
The study was conducted along a crooked 6.3 km long transect oriented approximately north–south (Figure 1)
running through several settlements. At the north and south ends are tidal creeks (Figure 1) whose salinity varies
seasonally. They are fresh in the monsoon period, but the salinity slowly rises during the dry season and by April
and May the creek water contains up to two thirds seawater (Bhuiyan et al., 2012).

84        In 2017, hydrological and lithological data were collected along the transect at high density (see Sect.

2.2.1) to a depth of 50 m in two field campaigns: one in the dry season (January–February) and one in the wet
season (July–August). To do so, groundwater observation wells were constructed, the ground was levelled, and
surface and groundwater were sampled. The hydrological data were used to establish (1) the present-day variation
in groundwater salinity (2) whether groundwater is affected by freshening or salinization (this entailed analysing
the cation exchange) and (3) the source water type recharging the groundwater (for this we analysed the isotopic
data). The lithological data were linked with the reconstructed geological history to determine the
palaeohydrogeological conditions in the study area from the Last Glacial Maximum (LGM) until the present day.






**Figure 1. Overview of the study area, indicating the transect and groundwater observation points, land use and current surface water channels. The land use types are based on satellite imagery, SRTM data, a soil map (FAO, 1959) and field observations. The location of the Department of Public Health Engineering (DPHE) borehole is indicated in the figure.**





### 2.2.1 Lithological drillings and groundwater observation wells

Groundwater observation wells were installed to collect lithological information and groundwater samples. In 2017, 34 tubes with filters at depths of between 6 and 46 m deep were installed at 20 locations. At 14 locations, a single tube was installed, at four other locations, a nest of two tubes was installed, and at the last four locations, a nest of three tubes was installed. Two groundwater observation wells (P16 and P17) were installed a year later, but as the sampling campaign had ended, no water samples are available for them.

The first drilling at each location was used to collect the lithological data to approximately 46 m depth (150 feet). The traditional "sludger" or "hand-flapper" method was used as drilling technique (Horneman et al., 2004). The drilling fluid was water from nearby surface water or tube wells, which was pumped out directly after installation by pumping the tube wells for at least 30 minutes or until EC (electrical conductivity) and temperature had stabilized. During the first drilling at each location, the sediment slurry was interpreted in the field every 1.524 m (5 feet). Additional lab analyses were performed on 47 sediment samples from the surface clay layer and at the filter depths. These samples were analysed for their grain size distribution with a Malvern Scirocco 2000, after pre-treatment to remove organic matter and carbonates and after peptizing the mud particles using a peptization fluid and ultrasound. The particles less than 8 µm were classed as clay, those between 8 and 63 µm as silt and those over 63 µm as sand (Konert and Vandenberghe, 1997).

The carbonate and organic matter contents of the samples were quantified by thermogravimetric analysis (TGA) using a Leco TGA-601. The percentage of organic matter was defined as the weight loss percentage between 150–550°C, corrected for structural water loss from clay by a factor of 0.07 times the fraction smaller than 8 µm (van Gaans et al., 2010; Hoogsteen et al., 2015). The carbonate content was determined as the percentage weight loss between 550 °C and 850 °C.

### 2.2.2 Elevation

Using a Topcon ES series total station (Topcon, Japan), surface elevation at the installed groundwater observation wells was measured relative to a zero benchmark (a concrete slab at nest 1). The elevation data were used to correlate the wells in terms of their water levels as measured at least two days after installation.

### 2.2.3 Hydrochemistry and isotopes

During two sampling campaigns, water samples were taken and analysed for anions (IC) and cations (ICP-MS); samples were also taken for tritium analysis, as well as for $\delta^2H$ and $\delta^{18}O$ analyses. For details, see Table 1. The groundwater samples in the groundwater observation wells was sampled at least a week after installation. The tube wells and groundwater observation wells were purged by pumping approximately three times the volume inside the tube. EC, temperature and pH were measured directly in the field using a HANNA HI 9829 (Hanna Instruments, USA). Alkalinity was determined by titration within 36 hours of sampling (Hach Company, USA).

**Table 1. Overview of chemical and isotope samplings.**

|  | Samples for IC and ICP-MS (N = 129) | Samples for $\delta^2H$ and $\delta^{18}O$ (N = 45) | Samples for tritium (N = 23) |
| --- | --- | --- | --- |



| Dry season sampling campaign (January and February 2017) | 26 groundwater observation wells, 68 tube wells, 10 freshwater ponds, 2 aquaculture ponds, 8 hand-auger borings | - | - |
|---|---|---|---|
| Wet season sampling campaign (July and August 2017) | 6 groundwater observation wells, 2 freshwater ponds, 3 aquaculture ponds, 3 hand-auger borings, 1 inundated field | 27 groundwater observation wells, 9 tube wells, 2 freshwater ponds, 3 aquaculture ponds, 3 hand-auger boring, 1 inundated field | 14 groundwater observation wells, 6 tube wells, 3 hand-auger borings |


131        For the IC and ICP-MS analyses, the water samples were stored in a 15 ml polyethylene tube after filtering

through a 0.45 µm membrane. Back in the Netherlands, aliquots were transferred to 1.5 ml glass vials with septum
caps for IC analysis. For the IC, the aliquots were diluted in accordance with their EC, which was used as an
approximation of their salinity. Below 2000 µS/cm the aliquots were not diluted (1:0), between 2000 and 4000
µS/cm the aliquots were diluted two times  (1:1), between 4000 and 10000 µS/cm the aliquots were diluted five
times (1:4), and above 10000 µS/cm the aliquots were diluted ten times (1:9). The remaining sample was spiked
by adding 100 µl of nitric acid ($HNO_3$), put on a shaker for 72 hours, and used for ICP-MS. For the ICP-MS, the
chloride concentrations were low enough to allow for direct measurement. However, samples 34, 128, 149, 150,
151, 164, 167, 169, 176, 178 showed matrix effects, so were remeasured after diluting five times. The samples for
isotope analysis ($\delta^2H$ and $\delta^{18}O$) were stored in 15 ml polyethylene tubes and analysed on a Thermo GasBench-II
coupled to a Delta-V advantage (Thermo Fisher Scientific, USA). Samples for tritium analysis were stored in
polyethylene 1 litre bottles and analysed according to NEN-EN-ISO 9698.
**2.3 Calculations and modelling**
**2.3.1 Variations in groundwater salinity**
To study salinity variation the water samples were classified on the basis of chloride concentration into four classes
(adjusted from Stuyfzand, 1993): fresh (chloride concentration <150 mg/l) brackish (chloride concentration 150–
1000 mg/l), brackish–saline (chloride concentration 1000–2500 mg/l) and saline (chloride concentration >2500
mg/l). We estimated the chloride values of observation wells P16 and P17 from their EC values: we assigned them
the chloride values of samples with similar EC values.
**2.3.2 Interpretation of isotopic composition**
The stable isotopic composition was interpreted using the mixing line between rainwater and seawater and the
Meteoric Water Lines of the two closest meteorological stations with isotopic data: Dhaka and Barisal (respectively
185 km and 125 km from the study area) (IAEA, 2017). For the mixing line between rain and seawater, the



weighted average rainwater composition was based on data from the Barisal meteorological station (IAEA, 2017)
and the seawater isotopic composition was based on Vienna Standard Mean Ocean Water (VSMOW).
**2.3.3 Cation exchange**
Evidence of cation exchange was assessed by calculating the amount of enrichment or depletion of the cations
compared to conservative mixing for each sample. Chloride was used as an indicator of the degree of conservative
mixing. The deviation from conservative mixing for compound $i$ (in meq/l) was calculated using the following
formulae, based on Griffioen (2003):
$iZ = i_{sample} - i_{conservative}$          (1)
with:
$i_{conservative} = i_{fresh} + \left(i_{sea} - i_{fresh}\right) \cdot \frac{(Cl_{sample} - Cl_{fresh})}{(Cl_{Sea} - Cl_{fresh})}$     (2)
where $i$ refers to the concentration in meq/l. Seawater is used for the saline end member $i_{sea}$. To calculate the Z-
values we used values from Ganges water (Sarin et al., 1989) and from pond water (this study) for the freshwater
end member $i_{fresh}$. We assumed that Ganges water has had a large influence on the study area for most of the
Holocene and that pond water might have influenced the groundwater recently. Z-values had to be negative or
positive for both freshwater end members to be accepted as being truly affected by hydrogeochemical processes.
To account for false positive or false negative Z-value due to errors in analysis, the Z-values also had to be larger
than the expected error for them to be interpreted as affected by hydrogeochemical processes. Like Griffioen
(2003), we assumed the expected error in the amount of exchange was 2.8%, based on a standard error in analysis
of 2% and standard propagation of error. The same formula was used to indicate whether sulfate had been depleted
by reduction or enriched by other sources.
**2.3.4 PHREEQC simulations of cation exchange**
For the interpretation of the hydrogeochemical processes that have occurred in each of the groundwater samples,
we needed to take account of site-specific conditions and site-specific hydrochemical processes; to do so, we used
the PHREEQC model code (Parkhurst and Appelo, 2013). Possible dissolution or precipitation of minerals was
assessed by calculating saturation indices for calcite, dolomite and gypsum, and the partial pressure for $CO_2$.
Additionally, cation exchange during salinization or freshening was simulated, to interpret the stage of salinization
or freshening for the samples. For salinization, a scenario was simulated in which seawater diluted 10 times
displaces Ganges water (Sarin et al., 1989). For freshening, two scenarios were simulated, because different cation
exchange patterns were expected for a) a scenario in which Ganges water displaces 10 times diluted sea water and
b) a scenario in which Ganges water displaces 100 times diluted sea water. The salinities assigned to the saline
water end members were based on the salinity levels of mostly less than 1/10th seawater detected in the
groundwater.

186       The Z-values of the samples were compared to the Z-values calculated in the PHREEQC scenarios.

Freshening or salinization was determined based on the NaZ value of the samples, with a positive NaZ value
indicating freshening and a negative NaZ value indicating salinization. Next, the simulated MgZ patterns in the
three PHREEQC scenarios were used to differentiate between the stages of freshening or salinization in the
groundwater samples.



The cation exchange processes were simulated using 1D reactive, advective/dispersive transport. The time steps were 1 year and the groundwater velocity was exactly one cell per time step, which makes the Courant number 1. The dispersivity was taken as half of the velocity, which resulted in a Peclet number of 2. The Courant and Peclet numbers were both within the boundaries of a stable model (Steefel & MacQuarrie 1996). The Cation Exchange Capacity (CEC) used in the model was based on the value calculated from the empirical variables for marine soils given by Van der Molen (1958):

$$CEC = 6.8 * A + 20.4 * B \tag{3}$$

where $A$ is the percentage of the particles smaller than 8 μm and $B$ is the organic matter percentage, based on values for the 26 sand samples taken in our study. Sulfate reduction and methane production were simulated by introducing $CH_2O$ in a zero-order reaction. For the salinizing scenario, $CH_2O$ was introduced at 0.1 millimoles per year. For the freshening scenario, 0.05 mmol per year were introduced. Based on the calculated saturation indices, calcite dissolution was simulated by keeping the calcite saturation index at 0.25 throughout the run, which is representative for marine water (Griffioen 2017).

## 3 Results

### 3.1 Regional hydrogeological reconstruction

The relevant Holocene sedimentary history of the researched upper 50 m of the subsurface starts in a landscape determined by conditions under the LGM. During the LGM, the sea level was much lower than it is today, leading to fresh conditions. The freshwater rivers eroded deeply incised valleys down to 120 m below the present-day land surface (BGS & DPHE, 2001; Hoque et al., 2014; Mukherjee et al., 2009). In the interfluvial areas, a Pleistocene palaeosol formed, characterized by oxidized sands (Umitsu, 1993; Burgess, 2010; Hoque et al., 2014). The LGM conditions in the study area are uncertain, as the area is near the edge of a possible palaeo-channel (Hoque et al., 2014; Goodbred et al., 2014), making it possible that the starting conditions for the Holocene sedimentation could be either a Pleistocene incised valley or a palaeosol.

The Holocene sedimentary history in southwestern Bangladesh can be divided into three distinct periods. First, at around 10–11 kyr BP (kilo years before present), a transgressive period started, when the sea level started to rise rapidly (Islam and Tooley, 1999). In combination with an increase in monsoon intensity (Goodbred and Kuehl, 2000b), this marks the start of a period with very rapid sedimentation (Goodbred and Kuehl, 2000a), leading to a transgressive sediment thickness of 20 to 50 m in nearby study sites (Sarkar et al., 2009; Ayers et al., 2016). The maximum inland location of the shoreline was either slightly south or slightly the north of our study area (Goodbred and Kuehl, 2000a; Shamsudduha and Uddin, 2007). During the transgression, sedimentary conditions are expected to have become more under the influence of marine salinity. In the second period – from 8 kyr BP – sea level rise slowed down, and at ~7 kyr BP the coast started to prograde (Goodbred and Kuehl, 2000a; Sarkar et al., 2009; Goodbred et al., 2014), which probably reduced the influence of marine water. Concomitantly, the monsoon intensity decreased, which caused the sedimentation rate to decline (Goodbred and Kuehl, 2000b; Sarkar et al., 2009). Finally, between 5 kyr and 2.5 kyr BP the Ganges moved eastwards (Allison et al., 2003; Goodbred and Kuehl, 2000a; Goodbred et al., 2003; Goodbred et al., 2014; Morgan and McIntire, 1959; Sarkar et al., 2009). The probable cause of the migration was a topographical gradient resulting from disproportional sedimentation by the Ganges in the west part of the delta (Goodbred et al., 2014), which reduced the supply of sand to the study area, resulting in smaller channels and a larger area of floodplain. In these tidal floodplains, silts and clays were

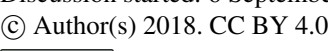


deposited during high water events (Allison et al., 2001), which is why clay overlies all of southwest Bangladesh
(BGS and DPHE, 2001; Sarkar et al., 2009; Ayers et al., 2016). Some small late Holocene channels depositing
fine sand were still present; sediments from such channels have been found in nearby study areas (Sarkar et al.,
2009; Ayers et al., 2016).
Even though the general trend since approximately 7 kyr BP has been progradation, there was a period
between 4.5 kyr and 2 kyr BP in which the sea level was higher than it is today (Gupta et al., 1974; Mathur et al.,
2004; Sarkar et al., 2009), which might have led to marine deposition at an elevation above present-day sea level
(Goodbred et al., 2003; Sarkar et al., 2009).
**3.2 Lithology**
The collected lithological data reveals a large variation in the thickness and organic matter content of the surface
clay layer. In the floodplains in the north at P9, P10, P16 and P17, and in the south at P15, the clay cover is
approximately 35 m thick and rich in organic matter (Figure 2, Table 3), whereas around the settlement in the
centre of the transect (henceforth referred to as the central settlement) it is 3–10 m thick and less rich in organic
matter (Figure 2, Table 3). From the middle of the transect towards the north and south, the clay cover becomes
gradually thicker (Figure 2). Under the clay cover is an aquifer composed of grey sands with carbonates, which
extends down to the end of all the drillings at 46 m depth (Figure 2, Table 3). This main aquifer contains some
small discontinuous organic-matter-rich clay layers (Figure 2, Table 3). Extrapolating from a log of the Department
of Public Health Engineering (DPHE) for a borehole located between P6 and P7 (Figure 2), it seems likely that
this sand layer extends to 110 m depth and is followed by a clay layer from 110 to 128 m depth and a second
aquifer down to a depth of 152 m. At N5, the surface clay layer was succeeded by a gravel bed at 10 m depth.
**3.3 Elevation**
The villages are at a different elevation than the rest of the study area. Compared with the benchmark, the elevation
of the groundwater observation wells in the villages (N2, P2, N3, P3, P5, P6, P7, P11, P12, P15) is  between 0.5
to 1.8 m higher, while the elevation in the agricultural fields (N1, P1, N5) and aquaculture ponds (P8, P14) is -0.6
to 0 m (Figure 2). The elevation was not measured at P9, P10, P16 or P17 but data from the SRTM and field
observations suggest that these areas are also relatively low.





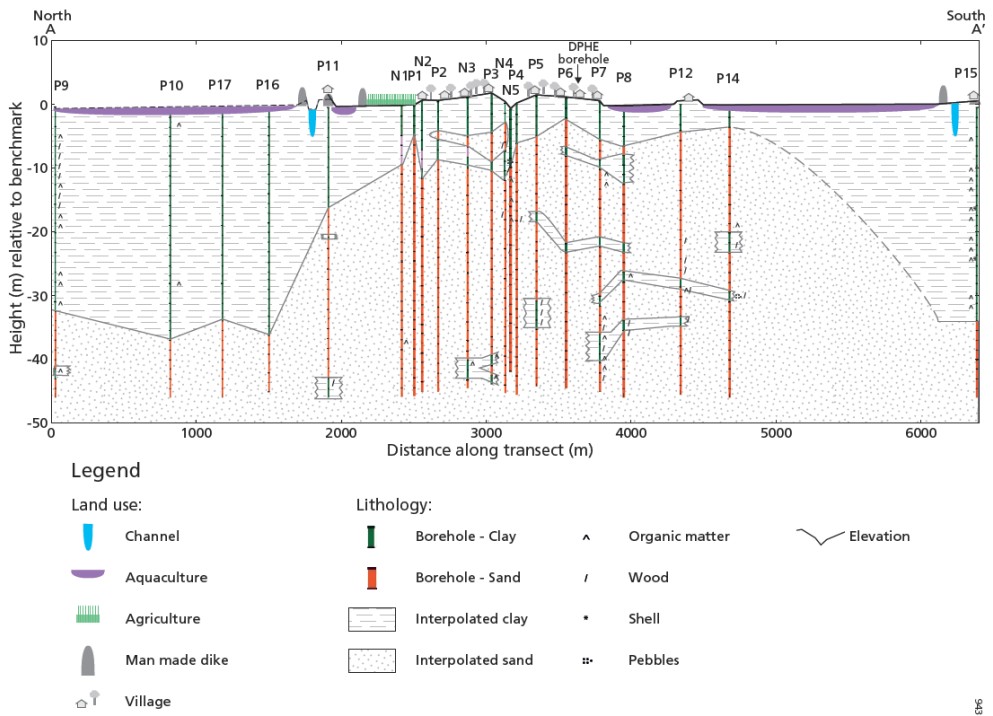


**Figure 2. Lithological data from the boreholes drilled in this study, and the interpolated sand and clay layers. A clear difference in thickness of the clay layer is visible between the palaeo floodplains at the north and south sides of the transect, and the palaeo channel in the middle of the transect.**

**3.4 Salinity**
The variation in surface water and groundwater salinity is shown in Figure 3. Surface and groundwater salinity are
discussed separately below.
**3.4.1 Surface water salinity**
The sampled surface water ponds can be divided into fresh and saline ponds. The ponds used for aquaculture
contain slightly less than 25% seawater and have a chloride concentration of around 4000 mg/l, and they are saline
throughout the year. The rainwater ponds in the settlements on higher land have a chloride concentration below
50 mg/l and are fresh throughout the year. In the wet season, additional surface water bodies are formed, when
many of the agricultural fields become flooded by the large amount of rain. The water in the flooded agricultural
field near N1 and P1 had a chloride concentration around 200 mg/l in July 2017 (Figure 3), which may be caused
by the dissolution of salts from the saline topsoil, as salt deposits are visible on the surface after the fields dry out
again in the post-monsoon period.
**3.4.2 Groundwater salinity**
The chloride concentrations of the groundwater samples vary between 18 mg/l and 4545 mg/l, which indicates that
the most saline groundwater samples contain somewhat less than 25% seawater. The salinity of the groundwater



in the first aquifer correlates well with surface elevation: higher areas are fresher than lower areas (Figure 3). At
the slightly higher central settlements, the groundwater is fresh to a depth of approximately 30 m. Below that
depth, the groundwater is brackish or brackish–saline. In the lower areas with a thick clayey top layer at the
northern and southern ends of the transect, the groundwater is brackish to brackish–saline (Figure 3). This was
also the case at P16 and P17, where the EC values were respectively 2.49 and 3.4 mS/cm, which –  based on the
chloride concentration of samples with similar EC values – corresponds with chloride values between 500 and
1000 mg/l.

282       In low areas with a thin clay cover, groundwater is saline–brackish under the agricultural areas and saline

under the aquaculture ponds. The groundwater salinity difference between these two land use types is  substantial:
groundwater chloride concentration under the aquaculture ponds exceeds 4000 mg/l, which is more than double
the groundwater chloride concentration under the agricultural areas (1000–2000 mg/l). The few samples taken to
the side of the transect show the same differences in groundwater salinity between these land use types. The salinity
of water samples taken a few metres below the surface from the clay is generally similar to that of the groundwater
in the sand aquifer immediately below, except for two clay water samples near the former creek in the middle of
the central settlement and one clay water sample from the agricultural fields north of the central settlement (Figure

290   3).

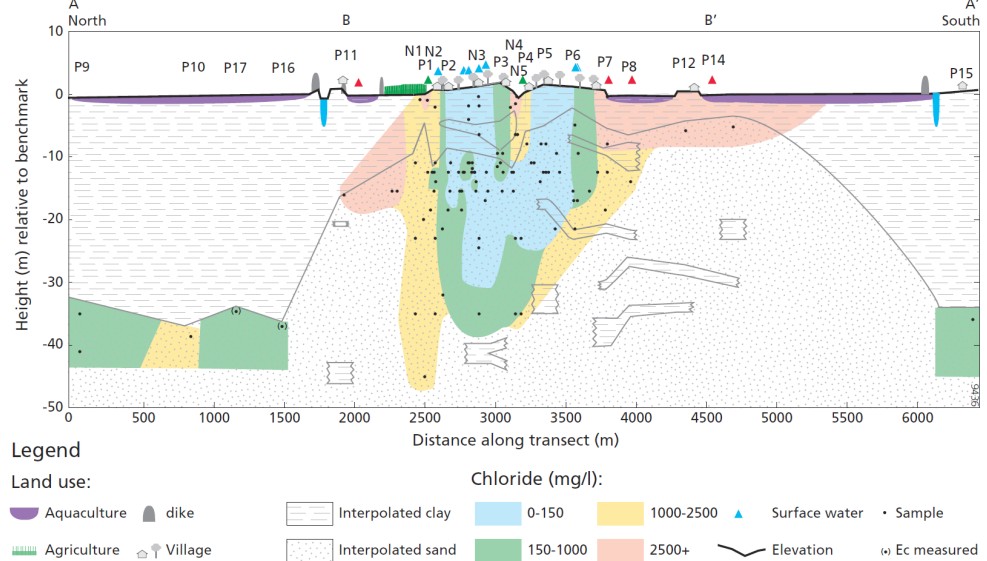

**Figure 3. Salinity of the sampled groundwater, of phreatic water sampled in clay just below the surface and of sampled**
**surface water bodies.**
**3.5 Stable isotopes**
The samples were divided into four source water classes (Figure 4). The first class consisted of samples with a
relatively light isotopic composition, similar to the weighted average rainwater. This indicates that direct rain
infiltration is the dominant source of this water. These samples were taken from surface water bodies in the wet
season. The shallow groundwater sample from the clay layer near N3 also falls in this class, revealing that direct
infiltration of rain occurs to some extent in the higher-lying area. Lastly, the groundwater at P7 and P8, and at P9,




P10 and P15 falls in this class, but since the samples were taken at great depths, or the isotopic composition of
overlying groundwater is different, it is unlikely this groundwater formed under present-day conditions (Figure 5).
The second class contains samples with a relatively heavy isotopic composition skewed to the right of the
MWLs (Figure 4). This indicates an effect of evaporation and mixing with seawater. Samples of this class were
taken at relatively shallow depths and close to surface water bodies (Figure 5), which suggests that the main source
of this water is water infiltrating from stagnant surface water.
The third class comprises samples with a relatively heavy isotopic composition located close to the MWLs
in Figure 4. In the study area, this class is visible in five groundwater samples taken just north of the central
settlements (Figure 5). The main water source of this class is unclear, but seems to be relatively heavy rain, with
limited evaporation or mixing with seawater.
The last and largest group is made up of intermediate weight samples without one clear water source type.
These samples could be a mix of different sources of water, such as rainwater, water from surface water bodies
and seawater. Some influence of surface water bodies is indicated by the small skew to the right from the MWLs.
Samples from this class were collected from both fresh and saline water under the thin clay layer (Figure 5).

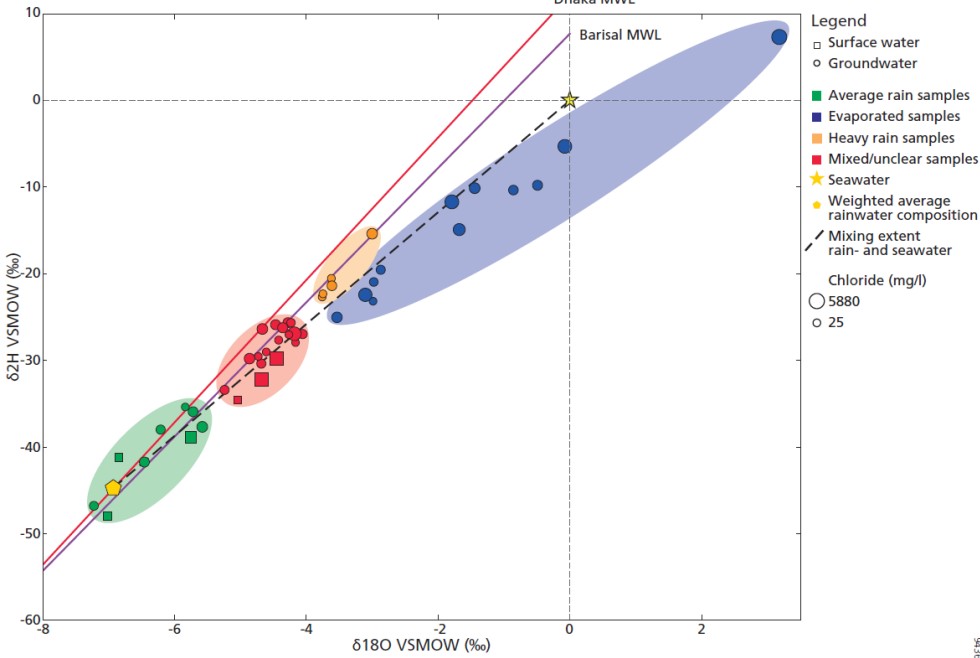


**Figure 4. Stable isotope content in the study area. Local Meteoric Water Lines based on monthly samples from the two closest stations Dhaka and Barisal at respectively 185 km and 125 km from the study area (IAEA, 2017). The mixing line between rainwater and seawater is based on the weighted average rainwater composition from Barisal (IAEA, 2017) and seawater VSMOW.**





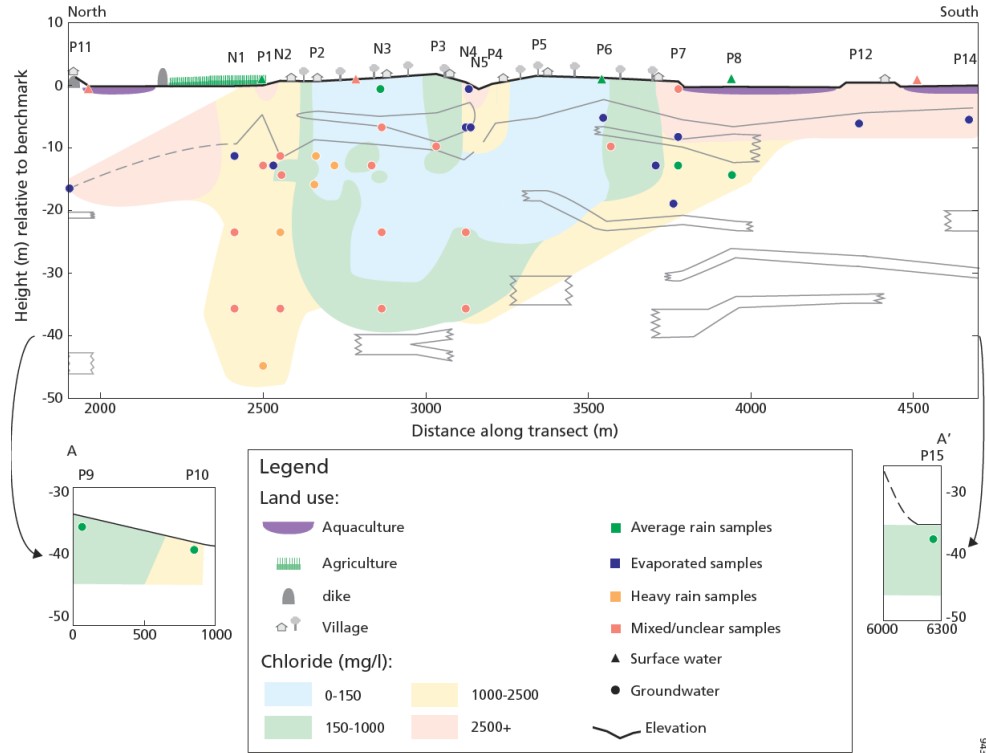

**Figure 5. Isotope group of the groundwater sampled in the cross section.**

**3.6 Tritium**

All 23 samples had a tritium concentration below the detection threshold of 1.64 Bq/l, and therefore tritium content could not be used to date the water. Seawater (<0.4 Bq/l), recent rainwater (0.5 Bq/l), or water older than 70 years would all have a tritium concentration below detection threshold, and might therefore be the source of the groundwater (IAEA, 2017).

**3.7 Redox conditions and saturation indices**

The conditions in the groundwater are reduced, with sulfate reduction and organic matter decay. Sulfate reduction is indicated by depleted sulfate concentrations compared to conservative mixing. Enrichment of $SO_4$ is observed only in the shallow saline clay near N4 and N5, possibly due to pyrite oxidation. Organic matter decay can be inferred from the partial pressure of $CO_2$ varying between $10^{-0.4}$ and $10^{-1.6}$ atm in most of the groundwater samples, which is high even for tropical conditions, but not unusual in Holocene coastal regions (Appelo and Postma, 2005; Griffioen et al., 2013).

Most of the groundwater samples from between 10 and 25 m deep are somewhat supersaturated for calcite, with saturation indices between 0 and 0.7, which is common for seawater-derived groundwater in coastal aquifers (Rezaei et al., 2005; Griffioen et al., 2013). Since the sediments contain carbonates (Appendix; Table 3), calcite, aragonite or dolomite is available for dissolution. The samples that were subsaturated for calcite were taken from the clay cover and the shallow aquifer (<10 m deep).





**3.8 PHREEQC simulations**
The Z-values of the groundwater samples were compared with the patterns of the Z-values during the salinization
and freshening scenarios (Figure 6). The patterns of the Z-values were used for the interpretation, as the exact Z-
values of the samples were expected to be lower than the Z-values in the model scenarios, since each sample is the
result of a specific, less extreme mix of end members. The CaZ, MgZ and NaZ of the groundwater samples were
plotted as points in the scenario whose Z-value patterns they best matched, with the X location determined by the
chloride concentration matching the values of the chloride in the model scenario. Samples with a chloride
concentration exceeding the chloride values in the scenario were plotted at the saline sides of the figures. Six cation
exchange (CE) groups were identified (Table 2).

**Table 2. Cation exchange groups identified.**

| Symbol CE group | NaZ value | MgZ value | Description |
|---|---|---|---|
| + | Positive | Positive | Freshening fresh (<200 mg Cl/l) |
| * | Positive | Negative | Freshening saline (<2000 mg Cl/l) |
| . | Neutral | Negative, neutral or positive | No cation exchange |
| - | Negative | Positive | Initial salinization |
| ~ | Negative | Neutral | Intermediate salinization |
| — | Negative | Negative | Late-stage salinization |


**3.8.1 Freshening**
The two freshening scenarios show different patterns for MgZ, because the saline water had a larger percentage of
Mg on the cation exchange complex than the fresh water (Appelo et al., 1987; Beekman and Appelo, 1991;
Griffioen, 2003). Consequently, the MgZ remained positive during the freshening in the freshwater freshening
scenario, while the MgZ became negative during the saline water freshening scenario. We therefore used the MgZ
values of the samples to differentiate between freshening freshwater samples (MgZ+) and freshening saline water
samples (MgZ-).
Freshening freshwater samples came from multiple locations in the shallow (<20 m deep) fresh
groundwater in the central settlement, which indicates that this fresh groundwater is likely formed by water
infiltrating from the surface. The freshening detected in the deep brackish samples at P9 and P15 is unlikely to be
caused by infiltrating fresh surface water, because of the thick impervious clay layer underlying the saline
aquaculture water (Figure 7). Like their isotopic composition, this suggests that this groundwater formed under
palaeohydrological conditions.
Freshening saline water was detected in three saline samples taken under the agricultural field north of
the central village (Figure 7). The freshening results either from infiltrating surface water, or from fresh water
flowing north, as suggested by head data measured in the groundwater observation wells.



### 3.8.2 Salinization

In the salinization scenario, the MgZ follows a clear sequence. The MgZ value rises initially, then falls until it becomes negative (Figure 6). The samples could therefore be divided into three salinization stages: initial salinization with a positive MgZ (- in Figure 7), intermediate salinization with a neutral MgZ (~ in Figure 7), and late-stage salinization with a negative MgZ (— in Figure 7). When initial, intermediate and late-stage salinizing groundwater is found in sequence, the direction of salinization can be interpreted. Groundwater at the salinization front is in the initial stage, behind it is intermediate salinizing groundwater, and finally late-stage salinizing groundwater is close to the source of the saline water.

Two clear salinization sequences are observed in the study area. One is south of the central settlement, with late-stage salinization close to the aquaculture ponds followed by intermediate salinization and initial salinization towards the fresh groundwater under the central village (Figure 7). This sequence indicates that the aquifer has been salinized from the surface of the lower areas. Another salinization sequence is visible at a former creek near N4 and N5 (Figure 7). Although the clay at the very top is freshening due to recent freshwater recharge, late-stage salinization is visible in the shallow aquifer, followed by intermediate and initial salinization in the brackish and fresh samples at around 12 m depth (Figure 7). This indicates that some of the groundwater has been salinized by water infiltrating  from this former creek.

Just north of the central settlement, the salinizing samples lack a clear sequence, but the different salinization stages can still be compared with each other to reveal differences in salinization processes. Under the agricultural fields near N1 and P1 the shallow brackish to fresh–brackish samples display initial salinization, whereas there is late-stage salinization in the samples from depths of  36.5 and 45.7 m. The samples from 36.5 m below the village on higher land again display initial salinization. This suggests that the deeper subsurface under the lower areas has been salinized for longer than the shallower subsurface.

The samples taken below the aquaculture ponds at P11, P12 and P14 display intermediate or late-stage salinization, suggesting that they have been salinized by water infiltrating from the aquaculture ponds above. However, as there are no samples close by with different salinization characteristics, it was not possible to determine the direction of this salinization front.

Some of the fresh and brackish water samples taken in the central settlement also display salinization. They have probably been salinized by some limited local saline water recharge, as the cation exchange characteristic in the fresh samples is sensitive to small changes in salinity. Near N3, the source of this local salinization is probably water infiltrating from the surrounding low areas, as the salinizing samples were taken close to the edge of the higher-lying area (Figure 1).





**Figure 6. Results of the freshwater freshening scenario (A), the saline water freshening scenario (B) and the salinization scenario (C), together with the samples that match each scenario according to the Z-values of their cations. The X location of the samples is determined by the chloride concentration matching the values of the chloride in the scenario. Samples with a chloride concentration exceeding the chloride values in the scenario are plotted at x= 250 or x=1750.**





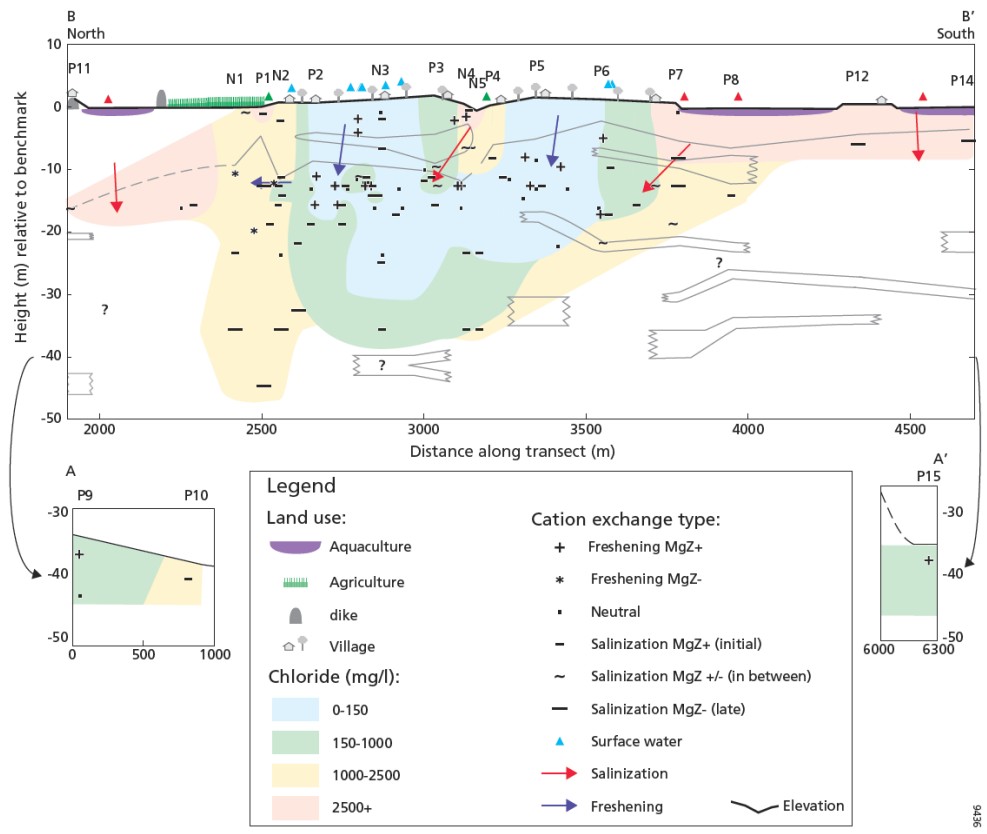

**Figure 7. Cation exchange types of the sampled groundwater and the phreatic water from the surface clay layer**

## 4 Discussion

### 4.1 Inferred hydrogeological evolution of the study area

Combining the field study results with the geological reconstruction based on literature enabled us to postulate a reconstruction of the evolution of the lithology and salinity distribution in the groundwater. This evolution is discussed below, considering the three main Holocene sedimentation phases described earlier plus an additional phase describing present-day processes.

### 4.1.1 Phase 1: Filling of an incised valley during Holocene transgression (10ka–7ka BP)

The Pleistocene palaeosol was not observed in the study area, which indicates either that a large incised valley was present in the Pleistocene, or that a Holocene channel later truncated the palaeosol (Hoque et al. 2014, Goodbred et al. 2014, Sarkar et al. 2009). This incised valley or truncated Holocene channel filled up rapidly during the transgression, when a rapid sea level rise combined with an increase in monsoon intensity led to accelerated sedimentation of channel sands (Figure 8a) (Islam and Tooley, 1999; Goodbred and Kuehl, 2000a; Goodbred and Kuehl, 2000b; Sarkar et al., 2009; Ayers et al., 2016). Therefore, the deeper part of the first aquifer sands must have been during the transgression.



### 4.1.2 Phase 2: Emergence of large lithological differences (c. 7 kyr– 5/2.5 kyr BP )

At some point in the early Holocene, channel meandering and associated sand deposition became limited to the middle of the transect (Figure 8b). This is indicated by the switch in sedimentary conditions from sand to clay deposition at approximately 30–35 m depth in the lower-lying areas at P9, P10 in the north, and P15 in the south. The unimodal, poorly sorted grain size distribution of this clay indicates that this switch probably occurred during the progradation (Sarkar et al., 2009). Throughout most of the rest of the Holocene, the areas near P9, P10 and P15 remained mangrove-forested tidal delta floodplains, while sand continued to be deposited by the Holocene channel around the present-day central village. The large difference in lithology indicates that the location of the channel was stable throughout the progradation. Possibly, mangrove vegetation in the floodplains prevented the channels from meandering due its ability to capture sediments up to the mean high water level (Furukawa et al., 1996; Auerbach et al., 2015) and to protect land against erosion (Van Santen et al., 2007; Kirwan et al., 2013).

This difference in lithology steered the influence of surface water on groundwater during the rest of the Holocene. The groundwater in the sandy aquifer in the middle of the transect continued to be influenced by the fresh surface water conditions, while the aquifers below the floodplains were much more isolated from surface influences by the thick clay layer. The groundwater under the thick clay layer must therefore be controlled by the hydrological conditions at the time of burial. Consequently, the thickness of the clay is the factor controlling the relative importance of palaeohydrological conditions for present-day groundwater salinity. We assume that during the progradation, the salinity at the surface decreased, as evidenced by the freshening cation exchange observed in the brackish groundwater near P9 and P15 (Figure 3). Additionally, the notion that this groundwater formed under different circumstances than the water close to the present-day central village is reinforced by the different isotopic composition of the groundwater below the thick clay layer compared to the isotopic compositions of the groundwater in the middle of the transect. This process of connate water sealing with subsequent limited influence has also been proposed by George (2013), Worland et al. (2015) and Ayers et al. (2016).

### 4.1.3 Phase 3: Clay deposition and formation of elevation differences (5/2.5 kyr–present)

#### a) Clay deposition

After the Ganges migrated eastwards between 5 kyr and 2.5 kyr BP, the areas that contained large sandy Holocene channels during the progradation also started to develop a clay cover (Figure 8c) (Goodbred and Kuehl, 2000a; Allison et al., 2003; Goodbred et al., 2003; Sarkar et al., 2009; Goodbred et al., 2014). The salinity during the deposition of the clay cover is not known, as it has been greatly affected by more recent freshening and salinization processes (see phase 4). Overall, however, more brackish conditions are likely to have prevailed from the moment the Ganges migrated eastwards, because the upstream supply of fresh water decreased. Possibly, the deeper groundwater at N1, P1 and N2 has been affected by salinization from this period, since its cation exchange characteristic indicates salinization at a later stage than the shallower samples (Figure 3).

Possibly, starting during the clay deposition in phase 3, salinization of the edges of the aquifer under the thick clay cover has occurred, due to density-driven flow from saline water infiltration in adjacent areas with a thin clay cover (Kooi et al., 2000). In the study area, however, brackish conditions are present at P16 and P17, indicating that density-driven flow has not affected the groundwater immediately below the thick clay layer. This does not mean that density-driven flow is not relevant in the study area. As salinization from density-driven flow





mainly consists of vertical convection cells, the resulting salinization can be expected to occur deeper than
immediately below the thick clay cover (Kooi et al., 2000; Smith and Turner, 2001).
**b) Formation of elevation differences**
While the clay cover was being deposited in phase 3, there was probably little difference in elevation. The observed
differences in elevation are thought to have come about after the clay deposition, as a result of differences in
autocompaction (Allen, 2000; Bird et al., 2004; Tornquist et al., 2008), which can lead to an inversion of surface
elevation (Vlam, 1942; van der Sluijs et al., 1965). The thick, organic-matter-rich clays under the floodplains are
likely to have been compacted more than the thinner clay cover on the former sand channels, which would account
for the present-day elevation differences of up to 1.5 m between the floodplains and the central former channel
area in the village on higher land. These elevation differences are similar to those observed in a comparable delta
areas in the Netherlands (Vlam, 1942; van der Sluijs et al., 1965).
The elevation differences in the middle part of the transect cannot be explained by autocompaction, as
here only small changes in lithology are observed. Instead, we hypothesize that they may result from erosion by
creeks at the edge of the higher areas. Evidence for this is provided by landforms that look like pathways of erosion
caused by meandering tidal creeks north of the central settlement, the tidal creek soils in the areas hypothesized to
be affected by erosion, and the distribution of tidal creeks and former tidal creeks in lower-lying areas overlain by
thin clay (FAO, 1959). Erosion by tidal creeks implies that the clays originally lay above the average tidal river
water level. A possible explanation is clay deposition during the higher sea level between 4.5 kyr and 2 kyr BP
(Gupta et al. 1974, Mathur et al. 2004, Sarkar et al. 2009). In a subsequent stage, the average water level of the
nearby tidal creeks dropped again, resulting in erosive channels.
**4.1.4 Phase 4: Emergence of groundwater salinity differences (present-day processes)**
**a) Higher areas: Freshening by rain and rainfed ponds**
The present-day small differences in elevation result in large differences in groundwater salinity, as the surface
elevation has determined whether freshening or salinization has occurred in the groundwater. In the higher-lying
areas, the conditions at the surface have mostly been fresh since the elevation differences came about, as the slight
elevation has prevented saline water flooding from tides and tidal surges. The fresh groundwater is recharged either
by direct infiltration of rainwater, and by infiltration of rainwater stored in man-made ponds. Direct rainwater
infiltration, which is a common formation process of freshwater lenses in elevated zones within saline areas (Goes
et al., 2009; de Louw et al., 2011; Stuyfzand, 1993; Walraevens et al., 2007), is indicated by the light isotopic
composition in the phreatic groundwater from the clay at N3.
Infiltration from the rainfed man-made ponds by humans could be a source of fresh groundwater, since
they contain fresh water year-round (Harvey et al., 2006; Sengupta et al., 2008). This enables infiltration of fresh
water in the dry season when the hydraulic head between the ponds and the groundwater is expected to be larger
than in the wet season. The evaporated isotopic composition of the groundwater at P6 suggests such infiltration of
pond water (Figure 5). However, the isotopic composition of the deeper fresh groundwater shows only a small
amount of evaporation (Figure 4, Figure 5), indicating that infiltration from the freshwater ponds is not the main
process responsible for the fresh groundwater – a conclusion reinforced by the usually very low permeability of



the pond bottoms (Sengupta et al., 2008), and the fact that construction of freshwater ponds occurred relatively
recently in geological terms (Kräzlin, 2000).
**b) Low areas: salinization by marine-influenced water**
Unlike the higher areas, the lower areas have often been flooded by tides or tidal surges. Recharge of this saline
surface water has been possible in the lower areas with a thin clay cover, where salinizing saline groundwater was
observed. This difference between salinization in the lower areas and freshening in the higher areas causes the
surface elevation to be the most important factor controlling the salinity of the groundwater in areas with a thin
clay cover. Since the salinization originates from the surface, there is already a large difference between higher
and lower areas in terms of salinity in the uppermost part of the subsurface, similar to the finding reported by
Fernández et al. (2010) for a delta area in Spain. This differs from the situation in Zeeland, the Netherlands, where
small fresh groundwater lenses are also present in low-lying areas (Goes et al., 2009; de Louw et al., 2011).
Erosion by tidal creeks also occurred in the middle of the central settlement, as visible at the former creek
location near N4, N5 and P4, which has salinized the existing fresh groundwater body, as evidenced by the chloride
concentrations (Figure 3) and the sequential salinization patterns (Figure 7) recharging on top of the fresh
groundwater.
Recently, aquifers under thin surface clay layers have become salinized by water infiltrating from the
overlying aquaculture ponds. This is observed at P11, P7 and P14, where the groundwater has a similar salinity to
the overlying aquaculture ponds and an isotopic composition that indicates large evaporation and mixing effects.
The salinization by saline aquaculture was detected only in the shallow groundwater underneath saline aquaculture
ponds in lower areas with thin surface clay layers. Slightly deeper, the salinization from the shrimp farms was no
longer observed; the samples from approximately 12 m deep at P7 and P8 are already much less saline (respectively
1230 and 1370 mg/l), and a totally different isotopic composition (Figure 3, Figure 5). Furthermore, no effect was
observed in aquifers under thick clay layers. This rather limited influence of saline aquaculture is not unexpected,
as saline aquaculture was only introduced in the study area approximately 30 years ago (Azad et al., 2009). It does
indicate that land use has become a controlling factor for the shallow groundwater salinity in areas with thin surface
clay layers. In the future, salinization from aquaculture ponds is expected to continue, and hence the extent of
salinization to increase. Since the low-lying aquaculture areas with thin surface clay layers are adjacent to the
higher areas, continued salinization from the aquaculture ponds could be a threat for the fresh groundwater under
the higher area.



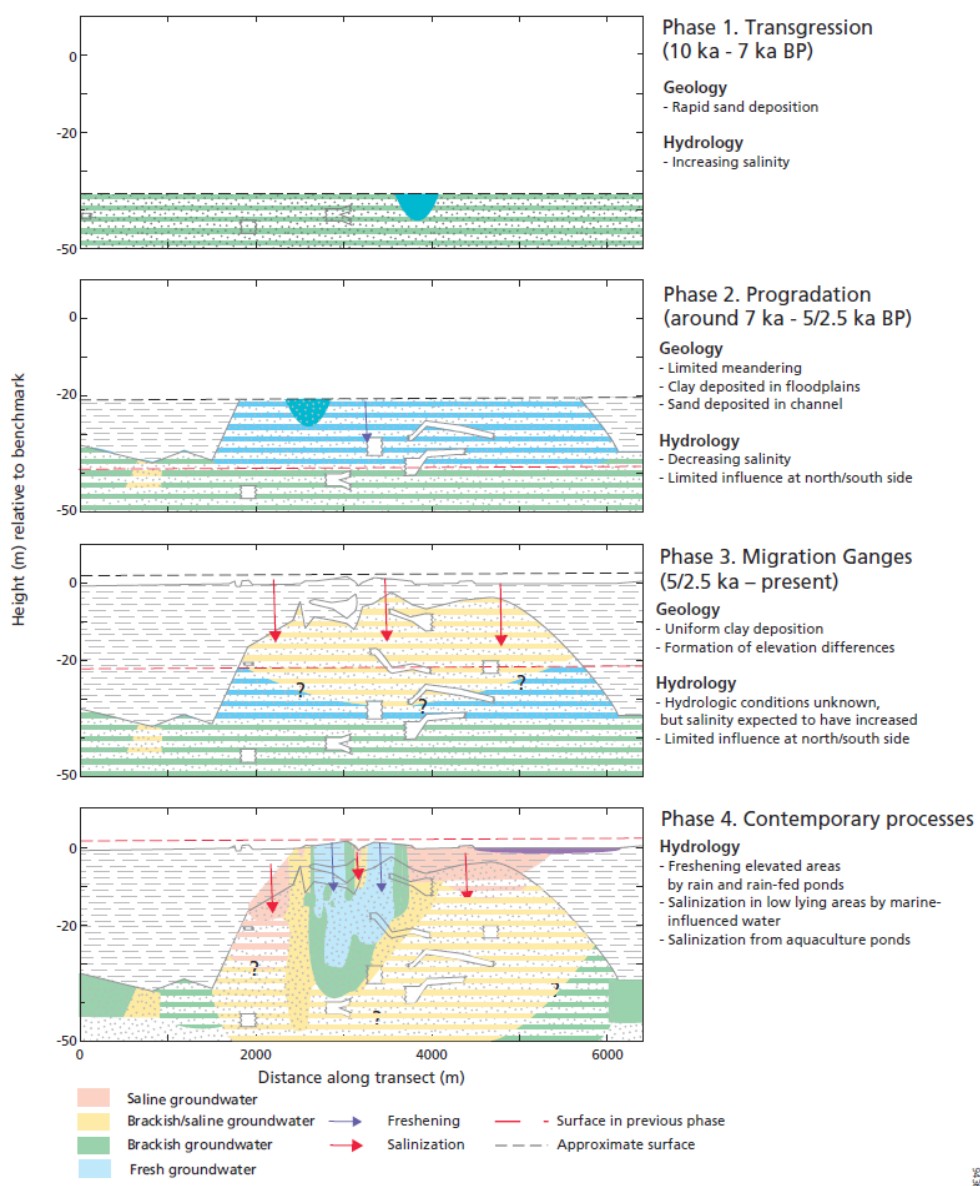

**Figure 8. Sediment build-up and associated freshening and salinization processes in the study area.**

**4.2 Reflection**

Above, we derived a hydrogeological evolution of a small area in southwestern Bangladesh, focusing on explaining the large variation in lithology and groundwater salinity which has often been reported in southwestern Bangladesh (BGS and DHPE, 2001; George, 2013; Worland et al., 2015; Ayers et al., 2016). Thanks to the high density of the sampling and the combination of salinity data with isotopic data and PHREEQC-interpreted cation exchange data, it was possible to establish clear patterns in groundwater salinity and to identify relevant hydrological processes and geographical and geological controls. Under the slightly higher area with a thin surface clay layer, a clear





pattern of fresh groundwater was identified, which is attributable to recharge by direct infiltration of rain or via
rainfed ponds. The presence of such fresh groundwater lenses in this region was postulated by Worland et al.
(2015) and Ayers et al. (2016), and the occurrence of fresh groundwater in elevated areas has been described in
other brackish or saline deltas (Stuyfzand, 1993; Walraevens et al., 2007; Goes et al., 2009; de Louw et al., 2011;
Santos et al. 2012), but to the best of our knowledge, these phenomena in southwestern Bangladesh have never
previously been reported in such detail. The fresh groundwater is bordered by brackish and brackish–saline
groundwater at greater depth under the higher area and in the direction of the adjacent lower areas, which probably
indicates mixing of fresh groundwater with recharged saline flood waters from the lower areas. Saline groundwater
is found only at relatively shallow depths below aquaculture ponds in areas with a thin surface clay layer and is
attributed to recharge from these present-day aquaculture ponds. Under thick surface clay layers in the lower areas,
brackish water is found; it is postulated to be controlled by palaeo salinity conditions at the time of sealing of the
sand aquifer. The importance of palaeo conditions for groundwater salinity in isolated parts of the subsurface in
this region has been mentioned by others (George, 2013; Worland et al., 2015; Ayers et al., 2016) and has been
described in other coastal zones (Sukhija et al., 1996; Groen et al., 2000; Post and Kooi, 2003; Sivan et al., 2005).
We postulate the hydrological processes described above and the resulting observed groundwater salinity variation
to be primarily steered by three geological and geographical controlling factors: clay cover thickness, relative
elevation and present-day land use.
We acknowledge our study has several limitations, and our interpretations should be seen as a conceptual
model to explain the observed spatial patterns of clay and sand deposits and of groundwater salinity. We did not
focus on quantifying recharge, discharge and flow rates, or the exact time scales of the hydrological processes. We
have been unable to discern comprehensive groundwater flow directions, aside from sketching some indicative
flow directions that would account for recharge, as we could not find evidence for locations and patterns of upward
groundwater flow and discharge. These upward groundwater flows are expected to be present in convection cells
caused by density-driven salinization (Kooi et al., 2000; Smith and Turner, 2001), and discharge is anticipated at
drainage points in the landscape which are thought to be present at the edge of higher areas and at the lowest points
in the landscape, i.e. the tidal rivers (Tóth, 1963). A possible next step would be to develop a numerical model to
further elucidate these flow processes, as well as estimates of recharge and discharge rates and time scales of the
described hydrological processes.
Despite these limitations, we contend that the identified controlling factors (clay cover thickness, relative
elevation and present-day land use) satisfactorily explain an appreciable part of the observed variation in
groundwater salinity variation in the larger southwestern Bangladesh region. Relative elevation and land use data
could provide a first estimate of the groundwater salinity in areas with a thin surface clay layer, while knowledge
of the palaeohydrogeological conditions seems to be necessary to understand and predict the groundwater salinity
in areas with a thick surface clay layer. A next step would be to test the validity of this hypothesis at regional scale.

*Data availability.* The data used in this study may be obtained by contacting the corresponding author.





**Appendix A**
**Table 3. The results of the sediment sample analyses. For the red TGA values no clear peak was identified, indicating**
**influence of TGA noise. For the orange TGA values the peak overlapped the border between the organic matter and**
**the carbonate temperatures.**

| Sample number | Sample location | Depth (m) | Grain size percentage | | | | | | Thermogravimetric analysis | |
|---|---|---|---|---|---|---|---|---|---|---|
| | | | % <8 µm | % 8–63 µm | % 63–126 µm | % 126–252 µm | % 252–502 µm | % >502 µm | % Organic matter (weight loss 150–550 °C) | % carbonates (weight loss 550–850 °C) |
| 1 | N1 | 5 | 15 | 57 | 21 | 5 | 2 | 1 | 1.51 | 2.43 |
| 2 | N1 | 12 | 2 | 11 | 19 | 52 | 14 | 1 | 0.49 | 2.02 |
| 3 | N1 | 24 | 0 | 7 | 18 | 51 | 23 | 2 | 0.45 | 2.01 |
| 4 | N1 | 37 | 16 | 18 | 13 | 30 | 19 | 4 | 1.34 | 1.90 |
| 5 | P1 | 5 | 40 | 56 | 2 | 0 | 0 | 2 | 0.40 | 1.13 |
| 6 | P1 | 14 | 1 | 9 | 39 | 44 | 6 | 1 | 0.33 | 2.58 |
| 7 | N2 | 5 | 14 | 63 | 18 | 4 | 1 | 0 | 1.26 | 2.79 |
| 8 | N2 | 12 | 5 | 18 | 32 | 40 | 5 | 0 | 0.57 | 2.33 |
| 9 | N2 | 24 | 1 | 10 | 15 | 48 | 19 | 7 | 0.41 | 2.04 |
| 10 | N2 | 37 | 0 | 10 | 19 | 46 | 20 | 5 | 0.55 | 2.15 |
| 11 | P2 | 8 | 6 | 28 | 33 | 24 | 6 | 2 | 0.76 | 2.69 |
| 12 | P2 | 12 | 0 | 9 | 18 | 54 | 16 | 2 | 0.56 | 2.15 |
| 13 | N3 | 5 | 41 | 56 | 2 | 0 | 0 | 2 | 0.29 | 1.84 |
| 14 | N3 | 8 | 2 | 12 | 27 | 48 | 10 | 1 | 0.58 | 2.02 |
| 15 | N3 | 11 | 15 | 49 | 17 | 14 | 4 | 2 | 2.81 | 2.57 |
| 16 | N3 | 24 | 1 | 8 | 13 | 44 | 29 | 5 | 0.30 | 1.87 |
| 17 | N3 | 37 | 0 | 6 | 8 | 32 | 40 | 14 | 0.30 | 1.96 |
| 18 | P3 | 5 | 30 | 63 | 4 | 1 | 0 | 2 | 0.51 | 1.63 |
| 19 | P3 | 9 | 0 | 9 | 29 | 44 | 14 | 3 | 0.56 | 2.23 |
| 20 | P3 | 24 | 0 | 11 | 21 | 35 | 25 | 9 | 1.05 | 1.97 |
| 21 | N4 | 3 | 22 | 68 | 8 | 1 | 0 | 1 | 0.98 | 3.23 |
| 22 | N4 | 8 | 1 | 9 | 16 | 40 | 27 | 7 | 4.86 | 1.56 |
| 23 | N4 | 9 | 11 | 46 | 22 | 14 | 6 | 1 | 1.75 | 3.37 |
| 24 | N4 | 24 | 0 | 9 | 27 | 48 | 13 | 3 | 0.39 | 2.50 |
| 25 | N4 | 37 | 0 | 6 | 15 | 50 | 25 | 3 | 0.32 | 2.14 |
| 26 | N5 | 9 | 15 | 43 | 16 | 17 | 7 | 2 | 1.86 | 2.99 |
| 27 | N5 | 12 | 0 | 2 | 5 | 30 | 55 | 9 | 0.41 | 1.33 |





| 28 | N5 | 24 | 0 | 6 | 17 | 58 | 17 | 2 | 0.36 | 2.18 |
| 29 | N5 | 37 | 0 | 6 | 15 | 44 | 26 | 8 | 0.39 | 2.14 |
| 30 | P4 | 3 | 24 | 67 | 6 | 1 | 1 | 2 | 1.18 | 3.63 |
| 31 | P4 | 9 | 0 | 10 | 22 | 48 | 17 | 3 | 0.46 | 1.70 |
| 32 | P5 | 5 | 22 | 65 | 9 | 2 | 1 | 2 | 1.18 | 4.26 |
| 33 | P5 | 9 | 2 | 16 | 26 | 38 | 16 | 3 | 0.36 | 1.82 |
| 34 | P6 | 3 | 14 | 55 | 22 | 6 | 2 | 1 | 0.42 | 3.73 |
| 35 | P6 | 6 | 0 | 5 | 20 | 49 | 21 | 4 | 0.44 | 1.97 |
| 36 | P6 | 9 | 7 | 23 | 25 | 36 | 8 | 2 | 1.48 | 3.00 |
| 37 | P6 | 23 | 0 | 7 | 16 | 52 | 24 | 2 | 0.31 | 1.82 |
| 38 | P6 | 24 | 17 | 55 | 17 | 6 | 3 | 1 | 3.07 | 2.63 |
| 39 | P7 | 9 | 2 | 25 | 44 | 20 | 7 | 3 | 0.83 | 2.89 |
| 40 | P7 | 14 | 1 | 11 | 28 | 39 | 16 | 5 | 1.55 | 2.07 |
| 41 | P8 | 5 | 12 | 53 | 26 | 7 | 2 | 0 | 1.56 | 3.07 |
| 42 | P8 | 15 | 0 | 7 | 31 | 43 | 16 | 3 | 0.54 | 2.15 |
| 43 | P8 | 37 | 1 | 5 | 7 | 33 | 49 | 5 | 4.65 | 1.50 |
| 44 | P9 | 3 | 32 | 61 | 4 | 0 | 0 | 2 | 1.94 | 1.73 |
| 45 | P9 | 15 | 26 | 65 | 7 | 1 | 1 | 1 | 3.03 | 3.53 |
| 46 | P9 | 32 | 23 | 65 | 10 | 1 | 1 | 0 | 4.86 | 1.65 |
| 47 | P9 | 37 | 1 | 13 | 39 | 39 | 6 | 2 | 0.54 | 2.92 |

*Author contributions.* PPS, KG and KMA wrote the proposal for this study's project. All authors contributed to the focus of the study and the design of the fieldwork campaigns. FLN carried out the fieldwork campaigns, analysed the results and wrote the manuscript with contributions from PPS and JG.

*Competing interests.* The authors declare that they have no conflict of interest.

*Acknowledgements.* This work is part of the Delta-MAR project funded by the Urbanizing Deltas of the World programme of NWO-WOTRO. We would like to thank all staff from the Delta-MAR office in Khulna for all their support during the fieldwork, notably Abir Delwaruzzaman for his continued efforts to make sure the fieldwork campaigns were successful. From Dhaka University, Atikul Islam and Pavel Khan are thanked for their field assistance. MSc students Frank van Broekhoven and Rebecca van Weesep (Utrecht University), and Aria Hamann (TU Delft) are acknowledged for helping collect and make a first interpretation of the field data. Dr Joy Burrough is acknowledged for editing a near-final version of the manuscript and Ton Markus is acknowledged for editing the figures.

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
