# Peer review of "Groundwater salinity variation in Upazila Assasuni"

_Hydrology and Earth System Sciences, 2018_

## Referee Comment (RC1) · Anonymous Referee #1 · 14 Nov 2018

Overview of Naus et al. (2018)

This study explores groundwater salinity variations in a village in southwestern Bangladesh using (1) geological evolution, (2) groundwater and lithological sampling, and (3) modeling. The paper is well written and logically describes the aim, methods, and findings of the groundwater investigation.

Major comments

–Figure 1 seems insufficient for the reader to fully understand the field site. An ad-
ditional satellite image would give the reader a better sense of the region. Perhaps consider a two-paneled figure with a satellite image adjacent to the site diagram. An additional figure in an appendix would also suffice.

–To connect back to the first paragraph of your introduction, I would be interested in how you could relate your findings back to drinking water availability. Is there enough freshwater to sustain the villages with drinking water? Is it likely that the freshwater recharge in high elevation regions will recharge at a pace similar to the salinization from the nearby low-lying regions and aquaculture? See line 520.

–One of the main conclusions is that elevation is correlated with groundwater salinity, where higher elevations are fresher than lower areas. This appears to be a poldered region surrounded by tidal creeks (as noted by the embankments in Fig. 2). The polder regions in SW Bangladesh are highly altered. The inhabitants have cleared forests, built embankments, compacted the landscaped by limiting flooding, dug ponds, and raised areas around their villages. There is a massive amount of sediment movement at the surface. Does this study area have ∼2 m natural variation in topography (partial answer in 4.1.3 b)? Or is the difference in elevation observed across the transect a recent development due to anthropogenic activity? If so, does that change how you would interpret the correlation between groundwater salinity and elevation? Could the salinity variations be more closely related to clay cap thickness?

Minor comments and additional questions

–What are the coordinates for the study area? This should be included in Fig. 1 or line 76.

–I'm a bit unsure how the "groundwater observation well" and "groundwater sample" data are different. Why include P16 and P17?

–Some of your groundwater tubes appear to be installed in the clay. In similar field environments, I have had difficulty extracting groundwater samples from clay layers,

especially using hand-flapper drilling methods. It always important to acknowledge the potential inaccuracies associated with this type of field environment. This type of tube installation can be a major source of error when dating or looking at cation exchange. Are you confident that you were able to evacuate the installation water, pump the tube, and extract groundwater despite the low permeability of the clay layer? It may be worth noting the potential limitations of the hand-flapper installation method.

–The variation in the clay cap thickness is substantial. Figure 3 shows the thickness varying from 5 to 35 m. The thickness of the cap has to play a major role in present-day surface water influence on groundwater salinity. Yet Figure 7 shows both freshening and salinization under the thickest region of the clay cap (near P9, P10, and P15). Can you better explain how the freshening and salinization stages can differentiate between current day processes (e.g. salinization due to aquaculture) and depositional processes (e.g. salinization due to evaporation of pore water during deposition of sediments 10 kyr BP)? Can you discuss any uncertainties associated with this methodology (without the use of dating), especially in an area with highly variable connate water? Most of these questions arose from the section 3.8.2. You do a nice job of answering some of these questions in your phased subsurface evolution in section 4.

Additional questions for discussion

–Do you see any seasonal changes in groundwater salinity associated with exchange with the tidal channels? If you see signatures of salinization and freshwater recharge in the thinned cap areas, how do you expect the groundwater signatures to compare near the tidal channels?

---

## Short Comment (SC1) · 11 Dec 2018

Overview of Naus et al. (2018)

This manuscript analyzes ground water salinity variations in southwestern Bangladesh using geological reconstruction, lithological sampling and modeling. The manuscript addresses a very important issue of water resource problem that the southwestern Bangladesh currently facing. According to the script ground water salinity mostly depends on the surface elevation. In the higher lying area with a thin surface clay layer always have a clear pattern of fresh ground water which stored by directly infiltration or via rainfed pond. The fresh ground water is bordered by brackish-saline water at greater depth under the higher area. On the other hand, lower area often flooded by tides and tidal surges and in the direction of the adjacent lower area saline ground water is found only at relatively shallow depth below aqua cultural ponds. So, here we can find that it is possible to clear the pattern in ground water salinity by using salinity data, PHREEQC-interpreted cation exchange data and to identify the hydrological process and geographical and geological control. Thickness, relative elevation and land used are the most common geographical controlling factor in which the ground water salinity depends.

However, there are some concern summarized below and I hope these comments could help the authors to improve the manuscript.

**Major comments**:

1) I am concerned about the sample preservation process use in the study since nothing was described in the paper. How were the sample preserved since you are running IC and ICP-MS back to the Netherland. I am thinking about NaCl concentration.

2) See line 520. To connect back to the first paragraph of your introduction, I would be interested in how you could relate your findings back to drinking water availability. Is there enough freshwater to sustain the villages with drinking water? Is it likely that the freshwater recharge in high elevation regions will recharge at a pace similar to the salinization from the nearby low-lying regions and aquaculture?

3) Your whole paper is based on the correlation between surface elevation and ground water salinity. Ground water salinity is higher when the elevation is lower and ground water salinity is lower when the elevation is higher (Fig 2). Thickness, relative elevation and land used are the most common geographical controlling factor in which the ground water salinity depends (line 546 and 546). But as far I know that 80 percent of the land mass of Bangladesh is made up of fertile alluvial lowland called Bangladesh plain. In plain area salinity basically depend on precipitation, infiltration and evapotranspiration (Yan,2015). So, from my point of view you should think about this too.

**Minor comments:**

▪ Line 63: Use it as a simple sentence because you are using too much "and" in one sentence and there should be no comma (and,) after and.

▪ Line 100: Two groundwater observation wells P16 and P17 were installed a year later but the sample campaign ended so no water is available for them. So, I don't understand why you installed P16 and P17.

- What is the difference between "ground water observation well" and "groundwater sample"?
- It is always important to acknowledge inaccuracy and error in the field work. Some of your tube is installed in the clay and it is very difficult to extract water from this condition. So, are you confident enough that there is no error in your field work? Because you didn't mention anything about that.
- In all over the manuscript there is lots of spacing problem. I can give you some example. In line 55: "Boun daries" should be "boundaries". In line 66: "Re constructed" should be "Reconstructed". And you can also check Line 216,217,232,244.
- Do you see any seasonal changes in groundwater salinity associated with exchange with the tidal channels? If you see signatures of salinization and freshwater recharge in the thinned cap areas, how do you expect the groundwater signatures to compare near the tidal channels?

**Figures and Table:**

- In Figure 1, there is no coordinates of the study area.
- Figure 3 shows the thickness varying from 5 to 35 m. The thickness of the cap has to play a major role in present-day surface water influence on groundwater salinity. Yet Figure 7 shows both freshening and salinization under the thickest region of the clay cap (near P9, P10, and P15). Can you better explain how the freshening and salinization stages can differentiate be- tween current day processes (e.g. salinization due to aquaculture) and depositional processes (e.g. salinization due to evaporation of pore water during deposition of sediments 10 kyr BP)?
- "Figure 1" seems like insufficient and difficult to understand. If you can add an additional satellite image, it will be helpful.
- In Figure 6, what is that dot line indicated.
- In Table 1, you should add more result for ICP-MS and IC test. Example, percentage of Al, As, Be, B etc.

**Reference:**

1) Yan, S. F., Yu, S. E., Wu, Y. B., Pan, D. F., She, D. L., & Ji, J. (2015). Seasonal Variations in Groundwater Level and Salinity in Coastal Plain of Eastern China Influenced by Climate. *Journal of Chemistry*, *2015*. https://doi.org/10.1155/2015/905190

2) Islam, K. R., & Weil, R. R. (2000). Land use effects on soil quality in a tropical forest ecosystem of Bangladesh. *Agriculture, Ecosystems and Environment*, *79*(1), 9–16. https://doi.org/10.1016/S0167-8809(99)00145-0

---

## Short Comment (SC2) · 11 Dec 2018

The comment was uploaded in the form of a supplement:
https://www.hydrol-earth-syst-sci-discuss.net/hess-2018-416/hess-2018-416-SC2-supplement.pdf

---

## Author Comment (AC1) · 5 Feb 2019

**Author's response to anonymous referee #1**

**1. Reviewer's comment:** This study explores groundwater salinity variations in a village in southwestern Bangladesh using (1) geological evolution, (2) groundwater and lithological sampling, and (3) modeling. The paper is well written and logically describes the aim, methods, and findings of the groundwater investigation.

**Author response**: We thank the referee for his time to review our paper.

**2. Reviewer's comment:** Figure 1 seems insufficient for the reader to fully understand the field site. An additional satellite image would give the reader a better sense of the region. Perhaps consider a two-paneled figure with a satellite image adjacent to the site diagram. An additional figure in an appendix would also suffice.

**Author response**: We agree that we can be more clear.

**Changes in manuscript**: We will add a figure with satellite imagery, either in the appendix or as a two-paneled figure. The following figure with satellite imagery will be added

[Figure]

**3. Reviewer's comment:** To connect back to the first paragraph of your introduction, I would be interested in how you could relate your findings back to drinking water availability. Is there enough freshwater to sustain the villages with drinking water? Is it likely that the freshwater recharge in high elevation regions will recharge at a pace similar to the salinization from the nearby low-lying regions and aquaculture? See line 520.

**Author response**: This is indeed an important question. Unfortunately, the exact fluxes of water are hard to estimate. It is difficult to assess the exact amount of water that is being extracted and used, as the fresh water is

used without regulation for both drinking/household water and irrigation water. Additionally, it is difficult to determine or estimate the recharge of fresh water, because of the (varying) clay cover thickness and the uncertain amount of recharge from fresh water ponds (article line 486-494).

We state that salinization from the aquaculture ponds could be a threat to the fresh groundwater, following the observation of salinization in the groundwater in the southern part of the village (Figure 7). The exact impact of salinization of low-lying regions on the fresh groundwater requires temporal measurements and modelling of groundwater flow.

For these reasons, we didn't aim to make an estimate of the fluxes and the sustainability of the fresh groundwater resource under present-day circumstances, which we acknowledge as a shortcoming in line 550. We see this as beyond the scope of this work, and as a topic for further research.

**Changes in manuscript**: No change

**4. Reviewer's comment:** One of the main conclusions is that elevation is correlated with groundwater salinity, where higher elevations are fresher than lower areas. This appears to be a poldered region surrounded by tidal creeks (as noted by the embankments in Fig. 2). The polder regions in SW Bangladesh are highly altered. The inhabitants have cleared forests, built embankments, compacted the landscaped by limiting flooding, dug ponds, and raised areas around their villages. There is a massive amount of sediment movement at the surface. Does this study area have _2 m natural variation in topography (partial answer in 4.1.3 b)? Or is the difference in elevation observed across the transect a recent development due to anthropogenic activity? If so, does that change how you would interpret the correlation between groundwater salinity and elevation? Could the salinity variations be more closely related to clay cap thickness?

**Author response**: We found a natural variation in topography of approximately 1.5m, based on our elevation measurements (par. 2.2.2).

It is true that a large anthropogenic influence is typical for the topography in the polders in Bangladesh. In our specific study area, we think that the main elevation differences are natural, for multiple reasons:

- The high area is very large, with a width of approximately 2 km (Figure 1), which is much larger than the anthropogenically created narrow high ridges next to creeks or next to roads that are present throughout most of the polders.
- The high area has a fluvial soil type, which suggests natural formation (FAO, 1959).
- This high lying area meanders naturally through the landscape, which also suggests a fluvial, natural origin.

"*Could the salinity variations be more closely related to clay cap thickness?*" When looking at Figure 3, we observe that the salinity of the groundwater is not only related to the clay cap thickness. In the low-lying areas directly surrounding the high-lying village (near N1, near P8 and near P14), the clay cap is similarly thin as below the high-area, but the groundwater salinity is saline instead of fresh. This indicates that clay thickness alone can't explain the salinity differences.

**Changes in manuscript**: No change

**5. Reviewer's comment:** What are the coordinates for the study area? This should be included in Fig. 1 or line 76.

**Changes in manuscript**: We added coordinates to Figure 1.

**6. Reviewer's comment:** I'm a bit unsure how the "groundwater observation well" and "groundwater sample" data are different. Why include P16 and P17?

**Author response:** 'Groundwater observation well' indicates a groundwater sample taken from a newly placed well (with a filter of 5 ft). A groundwater sample refers to any sample we took from the groundwater, including samples from household tubewells.

The installation of P16 and P17 had two goals:

1. They were used to better assess the lithological situation between P11 and P10.

2. They are used to determine whether density driven flow could lead to salinization of the water below the thick clay layer (see line 453).

**Changes in manuscript**: Clarified the type of tubewell: Added '*household*' in line 124 and in Table 1.

Clarified the use of P16 and P17: added '*to get more detailed lithological information between P11 and P10*' in line 100.

**7. Reviewer's comment:** Some of your groundwater tubes appear to be installed in the clay. In similar field environments, I have had difficulty extracting groundwater samples from clay layers, especially using hand-flapper drilling methods. It always important to acknowledge the potential inaccuracies associated with this type of field environment. This type of tube installation can be a major source of error when dating or looking at cation exchange. Are you confident that you were able to evacuate the installation water, pump the tube, and extract groundwater despite the low permeability of the clay layer? It may be worth noting the potential limitations of the hand-flapper installation method.

**Author response:** We are indeed not very clear in this regard. We didn't extract water from the clay by groundwater tubes, but we used the open auger boring method (De Goffau et al., 2012, see pictures below). We drilled a hole with a hand auger, without any drilling fluid. Then we waited for the hole to fill with water, which we sampled by inserting a sample bottle into the hole with a stick, which we pulled back out using a rope that was attached to the bottle. We called these samples retrieved from 'Hand-auger borings' in Table 1, but we didn't specifically described the field procedure of these samples.

[Figure]

[Figure]

[Figure]

**Changes in the manuscript**: Added a sentence in line 126 '*To sample porewater from the clay, we used the open auger boring method (De Goffau et al., 2012). We drilled a hole with a hand auger, without any drilling fluid. Then we waited for the hole to fill with water, which we sampled by inserting a sample bottle into the hole with a stick, which we pulled back out using a rope that was attached to the bottle.*' Changed the name of these samples from '*hand-auger borings*' to '*open auger borings*'

**8. Reviewer's comment:** The variation in the clay cap thickness is substantial. Figure 3 shows the thickness varying from 5 to 35 m. The thickness of the cap has to play a major role in present-day surface water influence on groundwater salinity. Yet Figure 7 shows both freshening and salinization under the thickest region of the clay cap (near P9, P10, and P15). Can you better explain how the freshening and salinization stages can differentiate between current day processes (e.g. salinization due to aquaculture) and depositional processes (e.g. salinization due to evaporation of pore water during deposition of sediments 10 kyr BP)? Can you discuss any uncertainties associated with this methodology (without the use of dating), especially in an area with highly variable connate water? Most of these questions arose from the section 3.8.2. You do a nice job of answering some of these questions in your phased subsurface evolution in section 4.

**Author response:** The stage of freshening or salinization alone cannot be used to differentiate between salinization by aquaculture or salinization by depositional processes. The salinization stages were used to determine the direction of salinization (see section 3.8.2).

The difference between present-day and depositional processes are, instead, postulated after combining the field results with the geological history described in literature, which we describe in section 4.1. In section 4.2, we acknowledge that our postulated evolution of the groundwater salinity is uncertain, and that it should be seen as a conceptual model.

**Changes in the manuscript:** Added a sentence in line 550: '*Without age dating, we can't determine the exact moment of salinization or freshening that has occurred.*'

**9. Referee comment:** Do you see any seasonal changes in groundwater salinity associated with exchange with the tidal channels? If you see signatures of salinization and freshwater recharge in the thinned cap areas, how do you expect the groundwater signatures to compare near the tidal channels?

**Author response:** We didn't look in detail at interaction of the aquifer with the tidal river, nor did we study temporal variety in salinity in the groundwater in great detail. As a consequence, we have no evidence for seasonal groundwater salinity near the tidal channel.

Without detailed and temporal measurements of water levels/heads in both the river and the aquifer, it is difficult to say what we expect near the tidal rivers. There are many factors at play that control exchange between the tidal channel and the aquifer, such as the depth of the tidal river, the thickness of the clay layer below the tidal river, possible clogging layers, changing water and salinity levels due to seasonality and tides (Bhuyian et al., 2012), possible salinity gradients in the tidal rivers and possible density driven flow.

At the northern tidal channel, not much exchange between the aquifer and the channel is expected, because of how narrow and shallow the channel has become.

**Changes in the manuscript:** No change

**References**

Bhuiyan, M. J. A. N. and Dutta, D.: Assessing impacts of sea level rise on river salinity in the Gorai river network, Bangladesh, Estuar. Coast. Shelf Sci., 96(1), 219–227, doi:10.1016/j.ecss.2011.11.005, 2012.

De Goffau, A., Van Leeuwen, T.C., Van den Ham, A., Doornewaard, G.J., Fraters, B.: Minerals Policy Monitoring Programme Report 2007–2010. Methods and Procedures. National Institute for Public Health and the Environment, Bilthoven, The Netherlands, RIVM Report 680717018, 2012

FAO: Soil Survey of the Ganges-Kobadak Area. ETAP Report no. 1071, Rome., 1959.

---

## Author Comment (AC2) · 5 Feb 2019

**Author's response to the short comment by Saif Rafi**

**1. Reviewer's comment:** This manuscript analyzes ground water salinity variations in southwestern Bangladesh using geological reconstruction, lithological sampling and modeling. The manuscript addresses a very important issue of water resource problem that the southwestern Bangladesh currently facing. According to the script ground water salinity mostly depends on the surface elevation. In the higher lying area with a thin surface clay layer always have a clear pattern of fresh ground water which stored by directly infiltration or via rainfed pond. The fresh ground water is bordered by brackish-saline water at greater depth under the higher area. On the other hand, lower area often flooded by tides and tidal surges and in the direction of the adjacent lower area saline ground water is found only at relatively shallow depth below aqua cultural ponds. So, here we can find that it is possible to clear the pattern in ground water salinity by using salinity data, PHREEQCinterpreted cation exchange data and to identify the hydrological process and geographical and geological control. Thickness, relative elevation and land used are the most common geographical controlling factor in which the ground water salinity depends.

However, there are some concern summarized below and I hope these comments could help the authors to improve the manuscript.

**Author's response:** We thank Saif Rafi for his time to review our paper.

**2. Reviewer's comment:** I am concerned about the sample preservation process use in the study since nothing was described in the paper. How were the sample preserved since you are running IC and ICP-MS back to the Netherland. I am thinking about NaCl concentration.

**Author's response:** After sampling, the samples were sealed tightly with the cap of the tube. In the field, it was not possible to actively cool the samples, but we kept the samples out of the sun. As soon as we got back to our overnight stay in the late afternoon, the samples were actively cooled, except when there was a power cut. During the transport to the Netherlands (by plane), the samples were put in a dark place, but they were not actively cooled. After arrival in the Netherlands, they were again cooled until analysis. The concentrations of major cations and chloride are not expected to have changed, as the cap was sealed tightly, and no leakage has been detected at all.

**Changes in manuscript:** We added a sentence: "*The samples were kept out of the sun and cooled as much as possible.*" (line 127)

**3. Reviewer's comment:** See line 520. To connect back to the first paragraph of your introduction, I would be interested in how you could relate your findings back to drinking water availability. Is there enough freshwater to sustain the villages with drinking water? Is it likely that the freshwater recharge in high elevation regions will recharge at a pace similar to the salinization from the nearby low-lying regions and aquaculture?

**Author's response:** (*same answer as to comment 3 of referee 1*) This is indeed an important question. Unfortunately, the exact fluxes of water are hard to estimate. It is difficult to assess the exact amount of water that is being extracted and used, as the fresh water is used without regulation for both drinking/household water and irrigation water. Additionally, it is difficult to determine or estimate the recharge of fresh water, because of the (varying) clay cover thickness and the uncertain amount of recharge from fresh water ponds (article line 486-494).

We state that salinization from the aquaculture ponds could be a threat to the fresh groundwater, following the observation of salinization in the groundwater in the southern part of the village (Figure 7). The exact impact of salinization of low-lying regions on the fresh groundwater requires temporal measurements and modelling of groundwater flow.

For these reasons, we didn't aim to make an estimate of the fluxes and the sustainability of the fresh groundwater resource under present-day circumstances, which we acknowledge as a shortcoming in line 550. We see this as beyond the scope of this work, and as a topic for further research.

**Changes in manuscript**: No change

**4. Reviewer's comment:** Your whole paper is based on the correlation between surface elevation and ground water salinity. Ground water salinity is higher when the elevation is lower and ground water salinity is lower when the elevation is higher (Fig 2). Thickness, relative elevation and land used are the most common geographical controlling factor in which the ground water salinity depends (line 546 and 546). But as far I know that 80 percent of the land mass of Bangladesh is made up of fertile alluvial lowland called Bangladesh plain. In plain area salinity basically depend on precipitation, infiltration and evapotranspiration (Yan,2015). So, from my point of view you should think about this too.

**Author's response:** Yan et al. (2015) researched the temporal groundwater salinity variation between the dry and wet season of shallow groundwater (5 m depth) in a very dynamic system, having a highly permeable soil and a high amount of infiltration. In Southwestern Bangladesh, the soil consists of clayey material with a much lower soil permeability, which leads to a much lower amount of infiltration. As a consequence, the system in Southwestern Bangladesh is much more stagnant than the system researched by Yan et al. (2015). The seasonal variation in precipitation and evapotranspiration, therefore, do not lead to changes in groundwater salinity. Additionally, we research the groundwater salinity at greater depths than Yan et al. (2015). The deeper groundwater is also less influenced by dynamics throughout the seasons. Instead, we describe that the differences in groundwater salinity form due to differences in surface conditions over longer time periods, which we describe in our article to depend on relative elevation and land use. We also acknowledge the importance of differences in infiltration: When there is a thick surface clay layer, the influences of the salinity conditions at the surface do not impact the groundwater salinity at larger depth.

**Changes in manuscript**: No change

**5. Reviewer's comment:** Line 63: Use it as a simple sentence because you are using too much "and" in one sentence and there should be no comma (and,) after and.

**Author's response:** The first 'and' separates 'freshening and salinization processes', while the second 'and' separates 'processes' from 'salinity'. We cannot remove either of them. The comma after the second 'and' is there because of the word 'therefore'. We think the sentence is grammatically correct.

**Changes in manuscript**: No change

**6. Reviewer's comment:** Line 100: Two groundwater observation wells P16 and P17 were installed a year later but the sample campaign ended so no water is available for them. So, I don't understand why you installed P16 and P17.

**Author's response:** (*same answer as to comment 6 of referee 1*). The installation of P16 and P17 had two goals:

1. They were used to better assess the lithological situation between P11 and P10.

2. They are used to determine whether density driven flow could lead to salinization of the water below the thick clay layer (see line 453).

**Changes in the manuscript**: added '*to get more detailed lithological information between P11 and P10*' in line 100.

**7. Reviewer's comment:** What is the difference between "ground water observation well" and "groundwater sample"?

**Author's response:** (*same answer as to comment 6 of referee 1*)

'Groundwater observation well' is a newly placed well (with a filter of 5 ft). A groundwater sample refers to any sample we took from the groundwater, including samples from groundwater observation wells and household tubewells.

**Changes in manuscript**: Clarified the type of tubewell: Added '*household'* in line 124 and in Table 1.

**8. Reviewer's comment:** It is always important to acknowledge inaccuracy and error in the field work. Some of your tube is installed in the clay and it is very difficult to extract water from this condition. So, are you confident enough that there is no error in your field work? Because you didn't mention anything about that.

**Author's response:** (*same answer as to comment 7 of referee 1*) We are indeed not very clear in this regard. We didn't extract water from the clay by groundwater tubes, but we used the open auger boring method (De Goffau et al., 2012, see pictures below). We drilled a hole with a hand auger, without any drilling fluid. Then we waited for the hole to fill with water, which we sampled by inserting a sample bottle into the hole with a stick, which we pulled back out using a rope that was attached to the bottle. We called these samples retrieved from 'Hand-auger borings' in Table 1, but we didn't specifically described the field procedure of these samples.

[Figure]

[Figure]

[Figure]

**Changes in the manuscript**: Added a sentence in line 126 '*To sample porewater from the clay, we used the open auger boring method (De Goffau et al., 2012). We drilled a hole with a hand auger, without any drilling fluid. Then we waited for the hole to fill with water, which we sampled by inserting a sample bottle into the hole with a*

*stick, which we pulled back out using a rope that was attached to the bottle.'* Changed the name of these samples from '*hand-auger borings*' to '*open auger borings*'

**9. Reviewer's comment:** In all over the manuscript there is lots of spacing problem. I can give you some example. In line 55: "Boun daries" should be "boundaries". In line 66: "Re constructed" should be "Reconstructed". And you can also check Line 216,217,232,244.

**Author's response:** Remarkably, we didn't see these spacing problems when we downloaded and viewed the pdf document on our computers, so we couldn't change anything. We have no explanation for the errors found in layout.

**10. Reviewer's comment:** Do you see any seasonal changes in groundwater salinity associated with exchange with the tidal channels? If you see signatures of salinization and freshwater recharge in the thinned cap areas, how do you expect the groundwater signatures to compare near the tidal channels?

**Author's response: (***same answer as to comment 9 of referee 1***)** We did neither look in detail at interaction of the aquifer with the tidal river nor did we study temporal variety in salinity in the groundwater in great detail. As a consequence, we have no evidence for seasonal groundwater salinity near the tidal channel.

Without detailed and temporal measurements of water levels/heads in both the river and the aquifer, it is difficult to say what one may expect near the tidal rivers. There are many factors at play that control exchange between the tidal channel and the aquifer, such as the depth of the tidal river, the thickness of the clay layer below the tidal river, possible clogging layers, changing water and salinity levels due to seasonality and tides (Bhuyian et al., 2012), possible salinity gradients in the tidal rivers and possible density driven flow.

At the northern tidal channel, not much exchange between the aquifer and the channel is expected, because of how narrow and shallow the channel has become.

**Changes in the manuscript:** No change

**11. Reviewer's comment:** In Figure 1, there is no coordinates of the study area.

**Changes in manuscript**: We added coordinates to Figure 1.

**12. Reviewer's comment:** Figure 3 shows the thickness varying from 5 to 35 m. The thickness of the cap has to play a major role in present-day surface water influence on groundwater salinity. Yet Figure 7 shows both freshening and salinization under the thickest region of the clay cap (near P9, P10, and P15). Can you better explain how the freshening and salinization stages can differentiate be- tween current day processes (e.g. salinization due to aquaculture) and depositional processes (e.g. salinization due to evaporation of pore water during deposition of sediments 10 kyr BP)?

**Author's response:** (*Same answer as to comment 8 of referee 1*). The stage of freshening or salinization alone cannot be used to differentiate between salinization by aquaculture or salinization by depositional processes. The salinization stages were used to determine the direction of salinization (see section 3.8.2).

The difference between present-day and depositional processes are, instead, postulated after combining the field results with the geological history described in literature, which we describe in section 4.1. In section 4.2, we

acknowledge that our postulated evolution of the groundwater salinity is uncertain, and that it should be seen as a conceptual model.

**Changes in the manuscript:** Added a sentence in line 550: '*Without age dating, we can't determine the exact moment of salinization or freshening that has occurred.*'

**13. Reviewer's comment:** "Figure 1" seems like insufficient and difficult to understand. If you can add an additional satellite image, it will be helpful.

**Author's response: (***Same answer as to comment 2 of referee 1***)** We agree that we can be more clear.

**Changes in manuscript**: We will add a figure with satellite imagery, either in the appendix or as a two-paneled figure. The following figure with satellite imagery will be added:

[Figure]

**14. Reviewer's comment:** In Figure 6, what is that dot line indicated.

**Author's response:** The dotted line indicates where Z-values are 0 (neutral).

**15. Reviewer's comment:** In Table 1, you should add more result for ICP-MS and IC test. Example, percentage of Al, As, Be, B etc.

**Author's response:** We will include a table with the relevant compounds for this study as supplementary material (Cl, Na, Ca, Mg, K, HCO3, SO4, Tritium, δ18O‰, δ2H‰)

**Changes in manuscript:** We will add a table as supplementary material.

**References**

Bhuiyan, M. J. A. N. and Dutta, D.: Assessing impacts of sea level rise on river salinity in the Gorai river network, Bangladesh, Estuar. Coast. Shelf Sci., 96(1), 219–227, doi:10.1016/j.ecss.2011.11.005, 2012.

De Goffau, A., Van Leeuwen, T.C., Van den Ham, A., Doornewaard, G.J., Fraters, B.: Minerals Policy Monitoring Programme Report 2007–2010. Methods and Procedures. National Institute for Public Health and the Environment, Bilthoven, The Netherlands, RIVM Report 680717018, 2012

Yan, S. F., Yu, S. E., Wu, Y. B., Pan, D. F., She, D. L., & Ji, J.: Seasonal Variations in Groundwater Level and Salinity in Coastal Plain of Eastern China Influenced by Climate. Journal of Chemistry, 2015. https://doi.org/10.1155/2015/905190, 2015